# Designing lead-free antiferroelectrics for energy storage

Bin Xu[1], Jorge Íñiguez[2] & L. Bellaiche[1]

Dielectric capacitors, although presenting faster charging/discharging rates and better stability compared with supercapacitors or batteries, are limited in applications due to their low energy density. Antiferroelectric (AFE) compounds, however, show great promise due to their atypical polarization-versus-electric field curves. Here we report our first-principles-based theoretical predictions that $Bi_{1-x}R_xFeO_3$ systems (R being a lanthanide, Nd in this work) can potentially allow high energy densities (100–150 J cm$^{-3}$) and efficiencies (80–88%) for electric fields that may be within the range of feasibility upon experimental advances (2–3 MV cm$^{-1}$). In addition, a simple model is derived to describe the energy density and efficiency of a general AFE material, providing a framework to assess the effect on the storage properties of variations in doping, electric field magnitude and direction, epitaxial strain, temperature and so on, which can facilitate future search of AFE materials for energy storage.

[1] Physics Department and Institute for Nanoscience and Engineering, University of Arkansas, Fayetteville, Arkansas 72701, USA. [2] Materials Research and Technology Department, Luxembourg Institute of Science and Technology (LIST), 5 Avenue des Hauts-Fourneaux, Esch/Alzette L-4362, Luxembourg. Correspondence and requests for materials should be addressed to B.X. (email: binxu@uark.edu) or to L.B. (email: laurent@uark.edu).

Today a rapidly increasing proportion of electricity comes from renewable sources. However, the intermittent nature of some of these sources (for example, wind, solar energy and so on[1]) poses the challenge of maintaining a balance between production and demand. Storing the energy temporarily, even for prolonged periods of time, is an indispensable requirement, but most of the available technologies—for example, compressed air, pumped hydro or even advanced batteries—lack the ability to respond fast enough (for example, in less than a second). In contrast, high-power energy storage systems, in particular electrostatic capacitors, are uniquely suited for effectively managing fluctuating energy sources[2,3].

Yet, storage applications using electrostatic dielectric capacitors are largely absent mainly due to their relatively low energy densities. Continued efforts are being devoted to find materials with high energy density, and antiferroelectrics (AFEs) are promising because of their characteristic polarization–electric field (P–E) double hysteresis loops schematized in Fig. 1a (ref. 4). For instance, a large energy density of ∼50 J cm$^{-3}$ has been achieved in AFE PbZrO$_3$-based films[5–7]. Attention is also paid to environmentally friendly lead-free systems[8–11], such as (Bi$_{1/2}$Na$_{1/2}$)$_{0.9118}$La$_{0.02}$Ba$_{0.0582}$(Ti$_{0.97}$Zr$_{0.03}$)O$_3$ (BNLBTZ) relaxor films that can reach energy densities of 154 J cm$^{-3}$ (ref. 10), comparable to good electrochemical supercapacitors[12]. However, the performance of these films relies on the coexistence of ferroelectric/antiferroelectric (FE/AFE) phases near the morphotropic phase boundary, which is sensitive to changes in composition and temperature. A simpler and more robust material—namely, AFE Hf$_x$Zr$_{1-x}$O$_2$—has been investigated by Park et al.[11] recently, but its energy density is about five times lower than that of BNLBTZ.

Here, we use first-principles-based simulation methods to investigate the energy-storage properties of a lead-free material, that is, Bi$_{1-x}$Nd$_x$FeO$_3$ (BNFO), which is representative of the family of rare-earth substituted BiFeO$_3$ (BFO) systems. BNFO and related compounds provide us with great flexibility to optimize their properties—especially the energy landscape that controls the transition between AFE and FE phases—by means of composition, E-field orientations and magnitude, as well as temperature and strain. This allows us to probe and understand the basic principles that lead to high energy density and efficiency of the AFE structure that is most common among perovskite oxides, that is, the orthorhombic Pnma phase. In particular, we find this system very promising due to its energy storage performance (at and around room temperature) and chemical simplicity: it is predicted to exhibit an energy density of 109–143 J cm$^{-3}$ for x > 0.5 with a maximum electric field of 2.81 MV cm$^{-1}$ (which has been experimentally achieved in BFO films without structural degradation under nanosecond high-voltage pulses[13]) and a good efficiency (68–88%). While minimizing leakage and consequently reaching high sustainable fields impose stringent demands for high-performance energy storage applications, techniques such as doping and advances in film fabrication are expected to provide a way forward. Further, we introduce a simple model that allows us to understand BNFO storage properties, and which should be useful in the search for other technologically-promising AFEs.

## Results

### Definition of energy density and efficiency.
Let us first concentrate on Fig. 1a, which shows the polarization-versus-electric field loop characteristic of AFEs. Such a loop involves an AFE state for small value of the E-field and a FE phase at high enough field. We will use $E_{up}$ to denote the critical field at which the AFE-to-FE transition occurs upon charging (increasing E), and

$E_{down}$ is the field associated with the FE-to-AFE transition upon discharging (decreasing E). The energy density W (green area in Fig. 1a) is defined from the discharging P(E) curve as $W = \int_{P_r}^{P_{max}} E dP$, where $P_r$ is the remanent polarization at zero field and $P_{max}$ is the polarization for the maximum applied field $E_{max}$. Note that $E_{max}$ should be smaller than the breakdown field, and that $P_r = 0$ if the AFE-to-FE transformation is perfectly reversible. Note also that $P_r$ and $P_{max}$ refer to the component of the total polarization along the direction of the applied field, and that we can have different energy densities depending on the field orientation. Similarly, we can define the energy density of the charging process $W'$ by calculating the above integral along the charging path. The difference between $W'$ and W quantifies the energy loss L (grey area in Fig. 1a), and the efficiency is defined as $\eta = W/(W + L)$.

### Solid solutions made of BiFeO$_3$ and orthoferrites.
Pure BFO has a high Curie temperature ($T_C \approx 1,100$ K) below which it adopts the FE R3c phase (Fig. 2b). Such a FE state possesses a large spontaneous polarization (about 90 $\mu$C cm$^{-2}$ at room temperature) arising from uniform polar distortions along the pseudo-cubic [111] direction, together with oxygen octahedral tiltings in anti-phase fashion about the same [111] axis (described by the $a^- a^- a^-$ Glazer notation[14]). Note that FE phases, unlike AFE states, are not ideal for energy storage due to the square shape of their hysteresis loops. Interestingly, an AFE Pnma phase (shown in Fig. 2a) has also been found in bulk BFO; this phase exhibits both anti-polar displacements of the A-site cations along the pseudo-cubic [110] direction and $a^- a^- c^+$ oxygen octahedral tiltings. However, this orthorhombic Pnma phase only exists at very high temperature (>1,100 K) in bulk BFO[15,16], which is not suitable for applications.

Such a limitation can be overcome by considering solid solutions of BFO with rare-earth orthoferrites. Previous studies have reported that the lanthanide substitution of Bi facilitates the stability of the Pnma phase, which becomes the equilibrium structure at room temperature for a moderate level of doping (for example, x > 0.3 for BNFO)[17–19] and gets increasingly stable with respect to the FE R3c state with larger rare-earth content. This is our main motivation to choose BNFO to optimize its energy storage capabilities at room temperature. For this purpose, we use a recently developed effective Hamiltonian approach (see Methods) that has been successful to reproduce the temperature-composition phase diagram of BNFO solid solutions[19] and to investigate hybrid improper FE effects (in BiFeO$_3$/NdFeO$_3$ short-period superlattices) as a function of temperature and under applied electric fields[20].

### P-E hysteresis curves.
We study BNFO compounds having different Nd compositions, but all known to adopt the Pnma phase at 300 K. We apply E-fields with four orientations—that is, the pseudo-cubic [001], [100], [110] and [111] directions—and a maximum magnitude, $E_{max}$, chosen to be 2.6 MV cm$^{-1}$; this is sufficient to cause the AFE–FE transition in all cases studied. Figure 3 depicts the resulting behaviour of the component of the polarization along the field direction as a function of the field magnitude, at room temperature. As shown in Fig. 3a, for increasing E along [001], the polarization increases smoothly within the AFE phase, and then abruptly jumps up at the AFE–FE transition. For all the investigated compositions, the [001]-oriented field results in the stabilization of the tetragonal T (P4mm) state sketched in Fig. 2c. This FE phase presents a large polarization pointing along [001] and no oxygen octahedral tilting; it is known to exist in BFO thin films under strong compressive epitaxial strain (>7% in magnitude)[21,22] and has

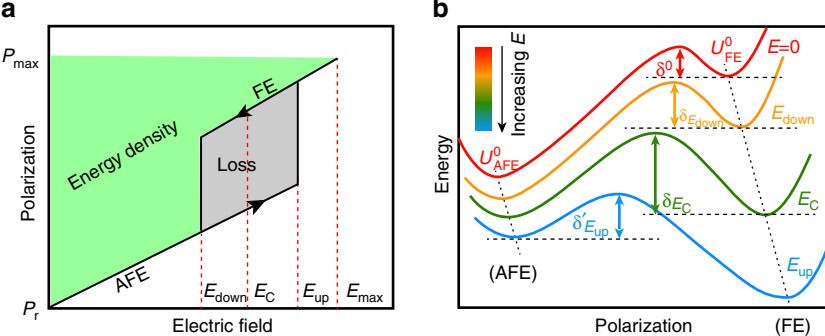

**Figure 1 | Schematic illustrations.** (**a**) The definition of stored energy density and energy loss from the typical polarization-versus-electric field double hysteresis loop of antiferroelectrics. The arrows indicate the charging and discharging processes. $E_{up}$ and $E_{down}$ denote the critical field at the AFE–FE and FE–AFE transitions, respectively. $E_C$ is the electric field at which the FE and AFE phases have precisely the same energy. (**b**) The energetic paths and barriers connecting the AFE and FE phases with increasing $E$-field.

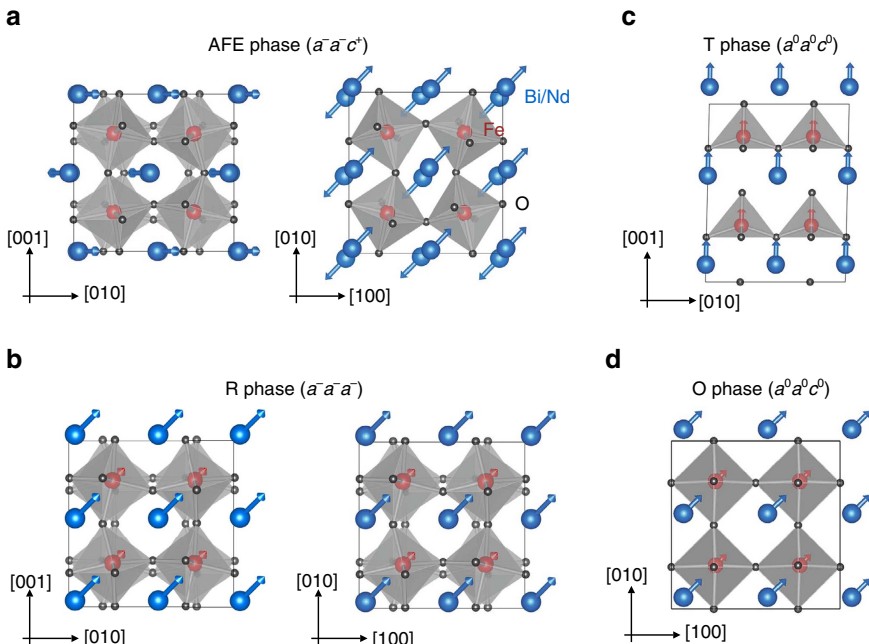

**Figure 2 | Relevant structures of $Bi_{1-x}Nd_xFeO_3$.** (**a**) The antiferroelectric orthorhombic *Pnma* phase (AFE phase), characterized by the anti-polar distortions along the pseudo-cubic [110] direction, and oxygen octahedral tiltings ($a^-a^-c^+$ in Glazer notations[14]). (**b**) The ferroelectric *R3c* phase (R phase), characterized by polar distortions and anti-phase tiltings about the [111] direction ($a^-a^-a^-$). (**c**) The ferroelectric tetragonal *P4mm* phase (T phase), characterized by polar distortions along the [001] direction and no tiltings ($a^0a^0c^0$). (**d**) The ferroelectric *Amm2* orthorhombic phase (O phase), characterized by polar distortions along the [110] direction and no tiltings ($a^0a^0c^0$). The VESTA code is used for the visualization[43]. Arrows represent local electric dipoles.

also been predicted to appear upon application of a large electric field to bulk BFO[23]. An intermediate phase may occur in a narrow $E$-field window (Supplementary Note 1), but is numerically found to have a small impact on energy density and efficiency. Note that intermediate phases have been experimentally reported for $Bi_{1-x}R_xFeO_3$ solid solutions in the compositional region that bridges the *R3c* and *Pnma* states[17,24–29].

The response to $E$-fields along the [100] direction, shown in Fig. 3b, shares many similarities to the case of the [001] field orientation. In particular, the final structure is also a T phase, the polarization now being along the [100] axis.

In contrast, for [110]-oriented fields (Fig. 3c) the AFE–FE transition leads to a *Cc* phase (Supplementary Table 1) with polarization along the pseudo-cubic [*uuv*] direction ($u > v$) and

anti-phase oxygen octahedral tiltings about [$u'u'v'$] ($u' > v'$). As $E$ increases, $u$ grows continuously while $v$, $u'$ and $v'$ all decrease. Upon two second-order transitions, $v$ and $v'$ first cancel and then $u'$ is annihilated, resulting in a final orthorhombic *Amm2* structure (O phase in Fig. 2d) with polarization along the [110] direction and with a $c/a$ ratio smaller than 1.

The fourth case studied corresponds to $E$-fields along [111], and the computed $P$–$E$ loops are shown in Fig. 3d. As consistent with previous studies, the AFE phase is found to transform into a complex intermediate structure and then into the FE *R3c* phase (Supplementary Table 1)[19].

It is worth noting that, for all four considered field orientations, when the field magnitude is reduced back to zero, the remanent polarization may differ slightly from zero. This is related with the fact that the FE phase becomes increasingly stable for small $x$

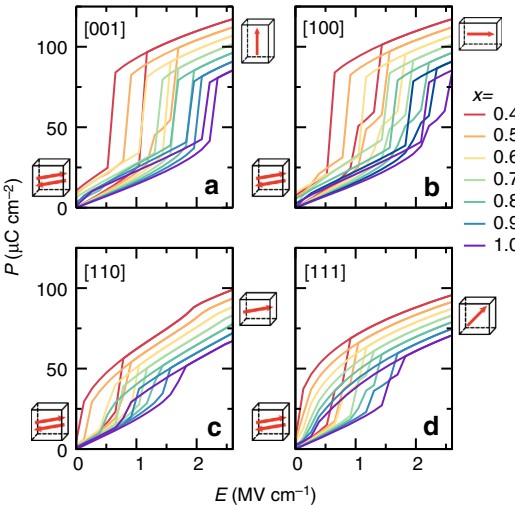

**Figure 3 | Calculated P–E hysteresis curves of Bi$_{1-x}$Nd$_x$FeO$_3$ solid solutions.** Nd composition ranges between 0.4 and 1.0, and four different electric field orientations are considered. (**a**) [001]. (**b**) [100]. (**c**) [110]. (**d**) [111]. The displayed polarization is the projected component of the total polarization along the direction of the applied E-field. The arrows inside the boxes (on the sides of each panel) schematize the direction of the long-range-ordered electric dipoles in the initial and final states. The different colours used for the solid lines denote compositions ranging from x = 0.4 to 1.0, as shown by the legend on the right.

(BFO-rich) compositions. The repeatability is investigated for several representative cases (Supplementary Note 2 and Supplementary Fig. 1), and the P–E curves of the first cycle are found to be reliable.

**Energy-storage performance of BNFO.** Let us now focus on the energy-storage performance associated with the P–E curves of Fig. 3. The symbols in Fig. 4a,b display the computed W and η, respectively, for $E_{max} = 2.6\,MV\,cm^{-1}$. Not surprisingly, the results for E-fields along [001] and [100] are essentially the same; therefore, we will omit the [100] case in the following. Regarding the magnitude of the energy density, the [001]-oriented fields give the best results, in particular for large x, while the poorest (though still remarkable) results correspond to the [111] orientation. In addition, W for the [110]-oriented field is comparable to the [001] result at x = 0.4, but has a rather weak dependence on composition.

The highest efficiency corresponds to the [110] fields and varies weakly with x. The efficiency for fields along [001] has a stronger composition dependence, and approaches the [110] result for x > 0.8. The smallest efficiencies correspond to the [111]-oriented fields.

Based on the predictions for a chosen $E_{max}$ of $2.6\,MV\,cm^{-1}$, we selected four representative cases and monitored the dependence of the energy storage properties on $E_{max}$: three concentrations (x = 0.5, 0.7, 1.0) were considered for field along [001] and a single composition (x = 0.5) for field along [110]. The results are shown in Fig. 5. For energy storage, it is general that the performance continuously improves with the magnitude of the applied field, particularly in what regards the energy density. Here, maximum field values $E_{max}$ up to $4.37\,MV\,cm^{-1}$ are considered (dotted line in Fig. 5), where this maximum corresponds to the intrinsic breakdown field estimated based on an empirical relation[30] that takes into account the experimental band gap of BFO[31]. Note that intrinsic breakdown should be taken as a theoretical upper limit, the actual breakdown being lower and depending on factors such as film quality, interface

with the electrodes and so on. Experience with related materials (for example, BFO) suggests experimentally feasible fields of $1–3\,MV\,cm^{-1}$; for example, a field of $2.81\,MV\,cm^{-1}$ has recently been applied to BFO films (54-nm-thick BFO film grown on a SrTiO$_3$ substrate) with nanosecond voltage pulses[13]. In addition to breakdown, energy storage has strict requirements on minimization of the leakage current. It was reported that rare-earth substitution in BFO can reduce the leakage current by two orders of magnitude as compared to that in pure BFO thin film[32]. Also note that Fig. 5a,b only include cases in which $E_{max} > E_{up}$, that is, for which the AFE–FE transition is possible under charging.

As expected from the definitions given above, both W and η increase with the maximum field value, emphasizing the key importance, for storage purposes, of being able to work with high fields. For any chosen $E_{max}$ with applied field along [001], a larger rare-earth content is found to yield better results, the compositional effect being more pronounced on the efficiency than on the energy density. For a fixed Nd composition of 0.5, W for the [110] case is comparable to that of [001] case for $E_{max}$ above $2\,MV\,cm^{-1}$. In contrast, η is generally much higher in the [110] case; for example, for x = 0.5 it is about 80% at $2\,MV\,cm^{-1}$ and goes above 90% for $E_{max} > 4\,MV\,cm^{-1}$. In fact, for $E_{max}$ larger than $2\,MV\,cm^{-1}$, the η obtained for the [110] orientation at x = 0.5 is similar to that of the [001] orientation but for x = 1 (pure NdFeO$_3$).

In Fig. 5b we also compare the energy density of BNFO with other previously reported top energy-storage materials—that is, lead-based[5,6,33–35] and lead-free[10,11] perovskites—for different experimentally applied $E_{max}$. We find that, taking BNFO with x = 1 and E-field along [001] as a reference, W is about three to four times that of (Pb,La)(Zr,Ti)O$_3$ (PLZT) films (for $E_{max} \sim 3\,MV\,cm^{-1}$)[5,6], five times larger than that of Hf$_x$Zr$_{1-x}$O$_2$ films (for $E_{max} \sim 4.5\,MV\,cm^{-1}$), and slightly higher than that of BNLBTZ (for $E_{max} \sim 3.5\,MV\,cm^{-1}$). The energy density of BNFO is also much higher than that of PVDF[33] ($27\,J\,cm^{-3}$ for $E_{max} \sim 8\,MV\,cm^{-1}$). In other words, the computed energy density of BNFO is comparable to that of BNLBTZ[10] and much higher than those of lead-based compounds, lead-free Hf$_x$Zr$_{1-x}$O$_2$ (HZO), and PVDF for similar or lower $E_{max}$; the efficiency of BNFO (for example, 90% for x = 1 with $E_{max} \sim 3.5\,MV\,cm^{-1}$) is also comparable to that of BNLBTZ (95% and decreasing with temperature) and much higher than that of HZO ($\sim 50\%$). Note also that, for lower $E_{max}$ (for example, $2–3\,MV\,cm^{-1}$) that is less challenging to be achieved in practice, W and η of BNFO are still considerably higher than those of PLZT and HZO (and are even larger than the W and η of PLZT and HZO obtained for $E_{max}$ just below $5\,MV\,cm^{-1}$). Moreover, Fig. 5b further shows that, up to the estimated intrinsic breakdown field of $E_{max} = 4.37\,MV\,cm^{-1}$, the energy density is predicted to be giant: it reaches values of 164, 191 and $213\,J\,cm^{-3}$ for x = 0.5, 0.7 and 1, respectively, with the E-field along [001], the corresponding efficiency being large as well (76, 88, 91%, respectively). Similarly, both W and η are very large, i.e., $161\,J\,cm^{-3}$ and 91%, for BNFO with x = 0.5 and E-field along [110] for the same $E_{max}$. These energy densities are comparable to that of supercapacitors (electrochemical capacitors), which is about $5\,Wh\,kg^{-1}$ ($\sim 125\,J\,cm^{-3}$ with the mass density of BNFO)[36]. BNFO materials therefore appear to be promising for energy storage purposes.

Note that, in addition to the direction and magnitude of the electric field, we also demonstrate that the energy density and efficiency of BNFO materials can be tuned by varying temperature (Supplementary Note 3) and epitaxial strain (Supplementary Note 4), in particular that the strong dependence of $E_{up}$ and $E_{down}$ with respect to strain can effectively

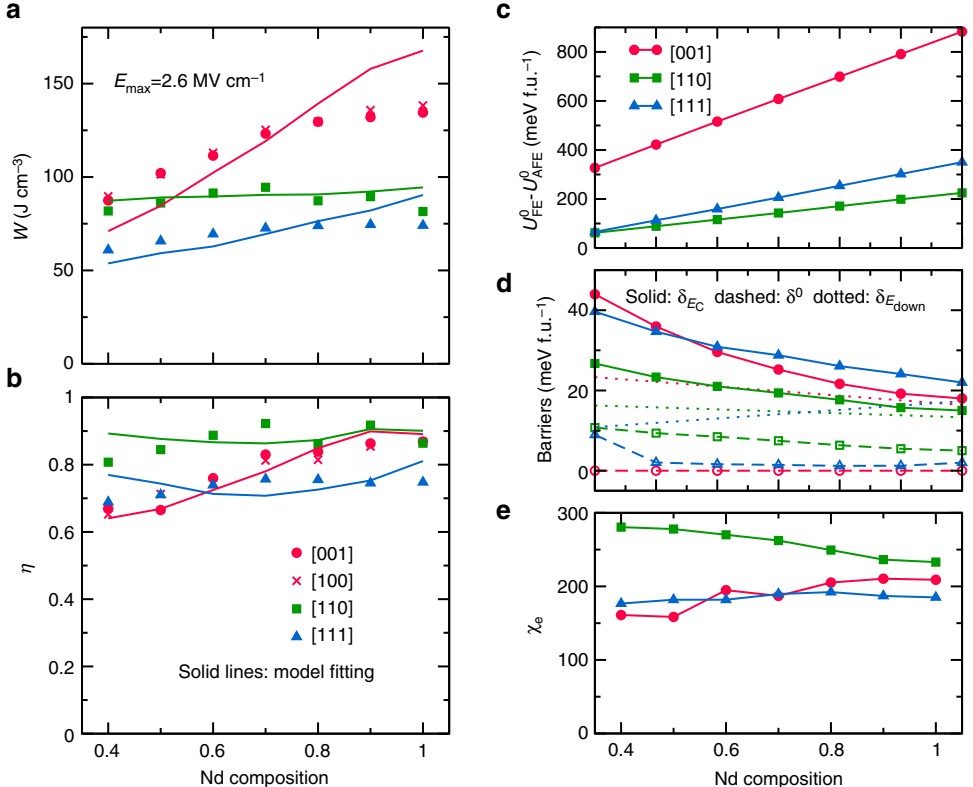

**Figure 4 | Energy-storage performance and model parameters. (a,b)** The calculated energy density and efficiency of $Bi_{1-x}Nd_xFeO_3$ solid solutions at various compositions ($x = 0.4$–$1.0$) and under various electric field orientations, with the maximum applied $E$-field ($E_{max}$) being $2.6\,MV\,cm^{-1}$ (larger than the AFE–FE transition field for all cases). The discrete data are calculated from the effective Hamiltonian simulations based on the predicted $P$–$E$ curves. The solid lines come from a least square fit of $W$ and $\eta$ to the simple model mentioned in the text (equations (6) and (7)) and Supplementary Note 6, with $\delta_{E_{down}}$ being the variable for each $E$ orientation with linear dependence on the composition. **(c–e)** Model parameters at various compositions and $E$-field orientations. **(c)** The zero-field energy difference between the AFE and FE phases. **(d)** The FE-to-AFE barrier at $E = 0$ ($\delta^0$, dashed lines), $E_{down}$ ($\delta_{E_{down}}$, dotted lines) and $E_C$ ($\delta_{E_C}$, solid lines). **(e)** The dielectric susceptibility.

accommodate different $E_{max}$ that can be achieved in experiment. On the other hand, using superlattices rather than disordered systems does not significantly affect energy storage performance, as discussed in Supplementary Note 5.

**Model and analysis**. One would like to have a simple and general model explaining the results in Figs 3 and 4 and, more generally, the relationship between the energy density and efficiency and the basic features of the energy surface of an AFE material (schematized in Fig. 1b). Let us begin by noting that, based on geometrical considerations about the areas defining $W$ and $L$ in Fig. 1a, we can deduce (Supplementary Note 6):

$$W = P_{FE}^0 E_{down} + \frac{1}{2}\epsilon_0 \chi_e E_{max}^2 \qquad (1)$$

$$\eta = \frac{P_{FE}^0 E_{down} + \frac{1}{2}\epsilon_0 \chi_e E_{max}^2}{P_{FE}^0 E_{up} + \frac{1}{2}\epsilon_0 \chi_e E_{max}^2} \qquad (2)$$

where $P_{FE}^0$ is the component of the polarization of the FE phase along the field direction as extrapolated to $E = 0$, and where we have assumed (for simplicity) that the dielectric response of both AFE and FE phases is identical (for a given field's direction) and given by $\chi_e$ (note that our effective Hamiltonian simulations indicate that $\chi_e$, in fact, decreases gradually with increasing $E$ in the FE phase). The FE-to-AFE and AFE-to-FE transitions are also assumed to be abrupt, based on the fact that these transitions were numerically found to occur in a narrow range of $E$-fields. These equations have straightforward interpretations. For

instance, $W$ and $\eta$ grow with $\chi_e$ and $E_{max}$, as expected. $W$ can also be optimized by having a FE phase as polar as possible (that is, with large $P_{FE}^0$) but which, at the same time, should be barely stable, so that the transition back to the AFE state occurs at a large $E_{down}$ field. In contrast, $\eta$ is optimized when the hysteresis is as small as possible ($E_{down} = E_{up}$).

Further, it is interesting to note that the critical electric fields and FE polarization can be expressed in terms of the parameters characterizing the relevant energy landscape of the AFE material (Fig. 1b). More specifically, let us consider (1) the relative stability of the AFE and FE phases as quantified by the energy difference $U_{FE}^0 - U_{AFE}^0$, where the '0' superscript indicates zero applied field; and (2) the energy barrier ($\delta$) that controls the FE-to-AFE transitions. As shown in Supplementary Note 6, if $E_C$ is the field at which $U_{FE} = U_{AFE}$, we can obtain

$$P_{FE}^0 = \left(U_{FE}^0 - U_{AFE}^0\right)/E_C \qquad (3)$$

and

$$E_{down} = \left(\frac{\delta_{E_{down}} - \delta^0}{\delta_{E_C} - \delta^0}\right)E_C \qquad (4)$$

$$E_{up} = \left(\frac{2\delta_{E_C} - \delta_{E_{down}} - \delta^0}{\delta_{E_C} - \delta^0}\right)E_C \qquad (5)$$

where $\delta^0$, $\delta_{E_{down}}$ and $\delta_{E_C}$ are the FE-to-AFE barrier at $E = 0$, $E_{down}$ and $E_C$, respectively (note that, as indicated in Supplementary Note 6, the energy barrier for the AFE-to-FE transition can be approximated as a function of the other parameters). Thus, the

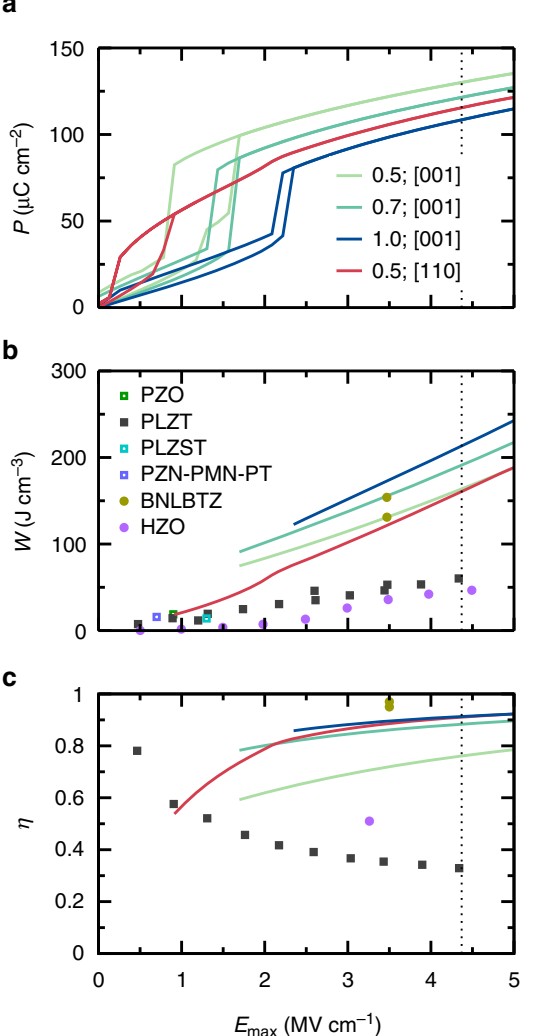

**Figure 5 | The computed energy storage performance of selected Bi$_{1-x}$Nd$_x$FeO$_3$ solid solutions.** (**a**) The P–E hysteresis curves. (**b**) The energy density as a function of the magnitude of the maximum applied electric field, with the discrete symbols representing the best available experimental data from different types of materials, that is, lead-based (PbZrO$_3$ (PZO)[34], PLZT[5,6,35,44], (Pb,La)(Zr,Sn,Ti)O$_3$ (PLZST)[33] and Pb(Zn$_{1/3}$Nb$_{2/3}$)O$_3$-Pb(Mg$_{1/3}$Nb$_{2/3}$)O$_3$-PbTiO$_3$ (PZN-PMN-PT)[33]), and lead-free (BNLBTZ[10] and HZO[11]) systems. (**c**) The efficiency as a function of the magnitude of the maximum applied electric field. The dotted vertical line denotes the estimated intrinsic breakdown field for BFO[30,31].

above equations can be rewritten to be:

$$W = \left( \frac{\delta_{E_{down}} - \delta^0}{\delta_{E_C} - \delta^0} \right) \left( U_{FE}^0 - U_{AFE}^0 \right) + \frac{1}{2}\epsilon_0 \chi_e E_{max}^2 \qquad (6)$$

$$\eta = \frac{\left( \frac{\delta_{E_{down}} - \delta^0}{\delta_{E_C} - \delta^0} \right) \left( U_{FE}^0 - U_{AFE}^0 \right) + \frac{1}{2}\epsilon_0 \chi_e E_{max}^2}{\left( \frac{2\delta_{E_C} - \delta_{E_{down}} - \delta^0}{\delta_{E_C} - \delta^0} \right) \left( U_{FE}^0 - U_{AFE}^0 \right) + \frac{1}{2}\epsilon_0 \chi_e E_{max}^2} \qquad (7)$$

Interestingly, the calculated W and η reported in Fig. 4a,b can be fitted to equations (6) and (7) with $\delta_{E_{down}}$—that is, the FE-to-AFE energy barrier at $E = E_{down}$—while the other parameters can be directly obtained from the employed effective Hamiltonian (Supplementary Note 7). $\delta_{E_{down}}$ thus quantifies the effective barrier to escape the FE state at a

particular temperature and for a particular E-field orientation, which is the key feature of the energy landscape of the compound. Figures 4a,b further report the resulting fitting curves, which are found to describe rather well the behaviours of the computed W and η (the free parameter $\delta_{E_{down}}$, together with the other coefficients of the simple model, are provided in Supplementary Table 2 and also reported in Fig. 4c–e). These good fits support the validity of our simple model for the relevant energy landscape, and its potential application to understand and analyse the energy-storage performance of AFE materials.

For instance, for E-field along [001], both W and η show considerable improvements with larger Nd content (Fig. 4a). According to the above equations and Fig. 4c–e, this can be understood as follows: (1) increasing x stabilizes the AFE phase and therefore $U_{FE}^0 - U_{AFE}^0$ increases; (2) $\delta_{E_C}$ decreases and gets close to $\delta^0$, a fact that is related to the reduction in the polarization (and stability) of the FE phase (see the correlation between the polarization and $\delta_{E_C}$ in Supplementary Fig. 5); and (3) the dielectric response increases with x.

As shown in Fig. 4c–e, for the case of a [110]-oriented E-field, we have a much smaller $U_{FE}^0 - U_{AFE}^0$, but the stronger dielectric response helps maintain a rather large energy density. The relatively small values of $\delta_{E_C}$ (Fig. 4d) also contribute to the high density and efficiency when the field is applied along [110]. Interestingly, our results also reveal that the weak dependence of W on x for fields along [110] mostly originates from a compensation effect: $U_{FE}^0 - U_{AFE}^0$ increases with Nd content while the dielectric response decreases.

Note that, as shown in Supplementary Notes 3 and 4, the proposed simple model can also be applied to explain other effects such as the dependence of W and η on temperature and epitaxial strain.

In summary, we conducted first-principles-based atomistic simulations that suggest that the family of rare-earth substituted BiFeO$_3$ compounds offers great opportunities for the optimization of energy storage in AFE capacitors. More specifically, we have studied the representative case of Bi$_{1-x}$Nd$_x$FeO$_3$ and obtained incredible performance indicators, such as energy densities of 150 J cm$^{-3}$ and efficiencies of 88%, for amenable applied electric fields of the order of 3 MV cm$^{-1}$. These compounds offer a wide variety of parameters to tune for optimized performance; here we have only explored in detail Nd content and the direction of the applied electric field, while epitaxial strain or the use of other rare-earth species (for example, Sm, Dy, Gd, La) are other promising possibilities. Finally, we have been able to explain the basic energy-storage features of these extraordinary materials on the basis of a phenomenological model that relies on a very simple, but in essence sufficient, description of the relevant energy landscape controlling the field-induced FE transition. The model analysis has shown that control of the dielectric response and relative stability of the involved phases is key to the optimization of the energy-storage properties, providing insights and a practical theory for further investigations of AFE-based capacitors.

We hope our results will motivate thorough experimental studies of these promising materials, which have so far been rarely considered in the context of AFE applications, in particular the Pnma phase which is the most common structure among perovskites. Indeed, our proposed Pb-free compounds appear as an appealing alternative for energy storage applications that are also environmentally friendly. In addition, we hope our results will motivate the search for experimental strategies to push up the breakdown fields in these compounds, and thus move towards the superior storage properties that our simulations predict.

## Methods

**MC simulation and effective Hamiltonian.** The finite-temperature properties of BNFO under electric field are presently predicted via the use of Monte-Carlo (MC) simulations, for which the total energy is provided by the effective Hamiltonian approach of ref. 19. The disordered $Bi_{1-x}Nd_xFeO_3$ solid solutions are simulated by $12 \times 12 \times 12$ supercells (containing 8,640 atoms), in which the Bi and Nd atoms are randomly distributed. For each electric field considered, we run 20,000 MC sweeps for equilibration and additional 20,000 MC sweeps to compute the statistical thermal averages. The calculated properties are found to be well converged.

Technically, the effective Hamiltonian of BNFO includes four types of degrees of freedom: (1) the local modes $\{\mathbf{u}_i\}$ centred on the $A$ sites (that is, on Bi or Nd ions), which are directly related to the local electric dipole[37,38]; (2) the homogeneous $\{\eta_H\}$ and inhomogeneous $\{\eta_I\}$ strain tensors[37,38]; (3) the pseudo-vectors $\{\boldsymbol{\omega}_i\}$ that characterize the oxygen octahedral tiltings[39]; and (4) the magnetic moments $\{\mathbf{m}_i\}$ of the Fe ions. (In all cases, the subscript $i$ labels unit cells in our simulation supercells.) In this Hamiltonian, a local quantity $\eta_{loc}(i)$ centred on the Fe-site $i$ is also introduced as $\eta_{loc}(i) = \frac{\delta R_{ionic}}{8} \sum_j \sigma_j$, where $\sigma_j$ (0 or 1) accounts for the presence of Bi or Nd ion at the $A$ site $j$ and the sum over $j$ runs over the eight nearest neighbours of Fe-site $i$ and where $\delta R_{ionic}$ represents the relative difference in ionic radius between Nd and Bi ions. $\eta_{loc}(i)$ is therefore different from zero if at least one of these eight $A$ sites are occupied by the Nd ions, while it vanishes for pure BFO.

The total energy of this effective Hamiltonian can be expressed as a sum of two terms

$$E_{tot} = E_{BFO}(\{\mathbf{u}_i\}, \{\eta_H\}, \{\eta_I\}, \{\boldsymbol{\omega}_i\}, \{\mathbf{m}_i\}) \\ + E_{alloy}(\{\mathbf{u}_i\}, \{\boldsymbol{\omega}_i\}, \{\mathbf{m}_i\}, \{\eta_{loc}\}) \qquad (8)$$

where $E_{BFO}$ is the effective Hamiltonian of pure BFO[16,40–42] and $E_{alloy}$ characterizes the effect of substituting Bi by Nd ions. Details about this method can be found in ref. 19 and references therein. Under an applied electric field, an additional term $-\sum_i \mathbf{p}_i \cdot \mathbf{E}_i$ is incorporated, where the local electric dipoles $\mathbf{p}_i$ are computed from the local modes $\{\mathbf{u}_i\}$ and effective charges $Z_i^*$.

Moreover, as the energy density depends on the absolute values of $E$ and $P$, we numerically find that, while the calculated $P$ is in good agreement with experiment, the simulated $E$-field is larger than the corresponding experimental field by an approximate factor of 23 (Supplementary Note 8), which is considered for all the energy densities.

**Data availability.** The data that support the findings of this study are available from the corresponding authors upon reasonable request.

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

## Acknowledgements

This work is financially supported by the Department of Energy, Office of Basic Energy Sciences, under contract ER-46612 (B.X.) and Air Force Office of Scientific Research under Grant No. FA9550-16-1-0065 (L.B.). We also thank FNR Luxembourg Grants FNR/P12/4853155/Kreisel CO-FERMAT (J.Í.) and INTER/MOBILITY/15/9890527 GREENOX (L.B. and J.Í.).

## Author contributions

L.B. and J.Í. conceived the project. B.X. conducted the study, carried out simulations, analysed the results and wrote the original manuscript. L.B. supervised the study. B.X., L.B. and J.Í. revised the manuscript. All authors contributed to the interpretation of the results.

## Additional information

**Competing interests:** The authors declare no competing financial interests.

**Publisher's note**: 

