## [Peer Review File · Nature Communications]

Reviewers' Comments:

Reviewer #1 (Remarks to the Author):

Xu et al. reported on a theoretical work where the antiferroelectric $\text{Bi}_{1-x}\text{Nd}_x\text{FeO}_3$ systems are predicted to yield excellent energy densities ($>250 \text{ J cm}^{-3}$) and efficiencies above 90%, using a first principle based computational method. The energy storage density of 250 J cm^{-3} and efficiency of 90 % should be the best result in this field, and certainly highly impressive and eye-catching result. Moreover, it is believed that the computational calculation method was also correctly conducted. However, there was a significant problem in their assertion; the authors assumed unrealistically high electric field and compare their results to the previous works on different materials. As a result, fair comparison cannot be made, and should have overestimate the experimentally achievable energy storage density from this suggested material. After considering the reasonable breakdown strength of $\text{Bi}_{1-x}\text{Nd}_x\text{FeO}_3$ system, the reviewer cannot recommend the publication of the current manuscript in Nature Communications. Detailed comments are as follows;

It is not easy to estimate the breakdown strength of materials, since the breakdown almost always occurs with extrinsic mechanisms related with defects such as oxygen vacancies. However, the intrinsic breakdown strength of materials is generally known to inversely proportional to their electric bandgaps. Wang suggested an empirical formula for the relationship between intrinsic breakdown strength and bandgap of semiconductors and insulators. [Wang, proceedings of 25th conference on microelectronics, 2006] Based on the equation suggested by Wang, the intrinsic breakdown strength of $\text{Bi}_{1-x}\text{Nd}_x\text{FeO}_3$ (Bandgap was $\sim 2.7 \text{ eV}$ for BiFeO_3 and $\sim 2.5 \text{ eV}$ for NdFeFO_3 , and 2.7 eV was taken.), PLZT ($\sim 3.4 \text{ eV}$), and HZO ($\sim 5.7 \text{ eV}$) were estimated as ~ 4.2 , ~ 8.4 , and 20.0 MV/cm , respectively. The maximum electric field used for energy storage were 2.6 , 3 , and 4.5 MV/cm for BiFeO_3 , PLZT, and HZO, respectively, which were ~ 62 , ~ 36 , and ~ 23 % of intrinsic breakdown strength. The authors took a value from a literature, but it was from an extremely thin ($\sim 4.6 \text{ nm}$) film for ferroelectric tunnel junction applications, and was measured using atomic force microscope. This value (62% of intrinsic breakdown field) was unreasonably high compared with those in other reports. Under such (unrealistic) assumption, the dielectric constant has significant impact on the calculated energy storage density ($\epsilon_0 \chi_0 E^2/2$). The χ_0 values in table I of SI were in the range of 161 (Nd content of 0.4 and (001) direction) – 281 (Nd content of 0.4 along (110) direction). The

energy storage density due to χ_0 values of 161 and 281 could be ~ 48 and ~ 84 J/cm³ at an external field of 2.6 MV/cm which was used in the manuscript. These values were surprisingly large compared with the total energy storage density, meaning that a significant portion of the calculated energy storage density originated from dielectric response. When the external electric field is decreased to be ~ 1.5 MV/cm (36% of intrinsic value as for the case of PLZT for fair comparison), the energy storage density due to χ_0 values of 161 and 281 decrease to ~ 16 (decrease by ~ 32) and ~ 28 (decrease by ~ 56) J/cm³. Moreover, when Nd content is higher than 70 %, the field induced phase transition might not occur within the maximum field of ~ 1.5 MV/cm, so the energy storage density might decrease even more seriously. Inversely, let's assume that ~ 62 % of intrinsic breakdown strength of HZO can be applied (12 MV/cm), then the energy storage density only from dielectric response (when χ_r is 30) could be as high as ~ 204 J/cm³, which is already much larger than the largest value reported in this manuscript. Please note that this large value does not count on the field induced ferroelectric polarization value. In practice, even for the high quality HZO film, only ~ 3 MV/cm can be reliably applied without significant breakdown.

To summarize, since the energy difference between polar and nonpolar phase increases with increasing Nd content, the energy storage density related with phase transition ($P_{0FE}E_{down}$) increases. However, when the practical breakdown strength is considered (far lower than the theoretical breakdown field), the field induced phase transition cannot be observed when Nd content is larger than 70%. For the case of the films with lower Nd content, the energy storage density due to dielectric response is rather high, but this value was too much overestimated due to the overestimated breakdown strength.

Reviewer #2 (Remarks to the Author):

In present study, the authors have proposed a lead-free antiferroelectric material for energy storage. For the material itself, the rare earth element substituted BiFeO₃ has been studied extensively. Most of them were focused on the crystal structure, ferroelectricity, and magnetism. It is of great interest that such a system could also play a potential role in the energy storage. The manuscript could be accepted for publication in NC. However, the authors need improve the manuscript by considering the following points.

1. Since the energy storage in antiferroelectric materials has been proposed for a while and many systems have been studied, could the authors provide a general picture on why Nd doped BiFeO₃ has such a superior performance? To do this, it is of great importance that the authors provide the simulation results of PbZrO₃ for comparison.
2. In a previous paper (PRB 85, 064119 (2012)), one of the authors has revealed a complex phase diagram and morphotropic region in La-doped BiFeO₃. In addition, it is noted that present simulation on Nd-doped BiFeO₃ is started from 60% BiFeO₃. So the question is what is the

exact phase diagram of $\text{Nd}_{1-x}\text{Bi}_x\text{FeO}_3$ and is the morphotropic region crucial for the performance in energy storage? Could it be considered in the simulation? As the authors described, compared with PZO, a three times increase of energy density was observed in BNLTZ and was believed related to the morphotropic region)?

3. Since Nd or La doped BiFeO_3 has been studied extensive, is there any experimental evidence (e.g. from P-E loop and dielectric measurement) that could support present theoretical results?

Reviewer #3 (Remarks to the Author):

This manuscript reported energy storage properties of lead-free antiferroelectric $\text{Bi}_{1-x}\text{Nd}_x\text{FeO}_3$ system predicted based on calculation using a first-principles-based computational method. Subject matter discussed and results reported in this current manuscript are of great interest to the audiences of the Nature Communications who are seeking solutions to high density energy storage applications. Technical presentation is mostly comprehensive and conclusions are largely supported by simulation and calculation results. However, there are a couple of critical issues must be addressed. Therefore, this manuscript is recommended for revision. Questions and suggestions are highlighted below.

Critical issues:

1. As it has been clearly stated in the Supplemental Information (SI) material, the framework of this manuscript is built on top of two approximations: (1) the dielectric response is independent of the magnitude of applied field and (2) the AFE to FE (and FE to AFE) transition is abrupt. It is okay to assume these approximations. But, they should be made clear in the main text. So that the audiences can make their own judgement as how reasonable those approximations are. The review believes that the first approximation is valid only small applied field. Under high field condition (in the FE region), the increase in polarization decrease with increasing applied field. Please provide a brief discussion in your revised manuscript.
2. All the calculation and simulation are conducted with unipolar hysteresis loops rather than the widely used bipolar hysteresis loops, contrary to your simulation results shown in Fig. 3a and SI Fig. 3 which had clearly shown that the remanent polarization is nonzero. This implies that if you had used a bipolar hysteresis loop instead of the unipolar loop, you would have gotten increased area enclosed by the hysteresis loops, i.e. reduced efficiency. Please estimate how much an error can potentially be introduced by using unipolar hysteresis instead of bipolar hysteresis loops in your revised manuscript.

Clarity issues:

3. Page 1 line 14: "...for experimentally achievable electric fields." Please be specific. This statement is confusing to the reviewer's point of view. You may give a value or a range for the applied field. Also, is "conventional electrostatic capacitors" the same as "dielectric capacitors"?
4. Page 4: Fig. 1b appears not referenced in the main text of the manuscript. If so, please consider of moving this figure to SI.
5. Page 6 line 4 from the bottom: "remnant" should be "remanent".
6. Page 7 line 2: "...the PE curve of Fig. 3." What are the solid lines in the Figure? Please explain.
7. Page 7 line 5: "...E_max value up to 10.8 MV/cm..." Are you sure that the approximations that this current manuscript based upon are still valid under this high electric field? If not, please consider of decreasing the upper limit of applied field in Figure 5. The efficiency data shown in Fig. 5 are questionable to the reviewer. If you had used bipolar hysteresis for the calculation. The charging curve will NOT start from zero. Rather, there will be a substantial enclosed area at near the AFE (near zero field) region. And the enclosed area increase with increasing maximum applied field. Please reference, Hu et al., "Temperature-dependent energy storage properties of antiferroelectric PLZT thin films," Appl. Phys. Lett. 104 (2014) 263902 for experimental data.
8. Page 8 line 1: "...both W and η increase with the maximum field value..." see above comments.
9. Page 8 line 5: "For a fixed...pure NFO). Please recheck the validity of this discussion.
10. Page 9 line 6 from bottom: What does "energy surface" refer to here? Please explain.

Reviewer #1 (Remarks to the Author):

1. “Xu et al. reported on a theoretical work where the antiferroelectric $\text{Bi}_{1-x}\text{Nd}_x\text{FeO}_3$ systems are predicted to yield excellent energy densities ($>250 \text{ J cm}^{-3}$) and efficiencies above 90%, using a first principle based computational method. The energy storage density of 250 J cm^{-3} and efficiency of 90 % should be the best result in this field, and certainly highly impressive and eye-catching result. Moreover, it is believed that the computational calculation method was also correctly conducted. However, there was a significant problem in their assertion; the authors assumed unrealistically high electric field and compare their results to the previous works on different materials. As a result, fair comparison cannot be made, and should have overestimate the experimentally achievable energy storage density from this suggested material. After considering the reasonable breakdown strength of $\text{Bi}_{1-x}\text{Nd}_x\text{FeO}_3$ system, the reviewer cannot recommend the publication of the current manuscript in Nature Communications. Detailed comments are as follows”

Answer: We thank the reviewer for his/her insightful comments. However, even if it is currently challenging to reach very high electric field due to breakdown, we would like to emphasize that a field of 6.5 MV/cm has been experimentally achieved in BiFeO_3 films (please see ACS Nano 7, 5385–5390 (2013)). Moreover, as we show below in the answer of comment #2 of Reviewer #1, the energy density and efficiency of $\text{Bi}_{1-x}\text{Nd}_x\text{FeO}_3$ is still promising at low fields, as compared to HZO and PLZT.

Furthermore, the objective of the manuscript is not only to report the promising storage properties of the specific compound $\text{Bi}_{1-x}\text{Nd}_x\text{FeO}_3$, but also to emphasize the very promising possibility of using antipolar materials with the *Pnma* perovskite structure (that is, the most abundant type of perovskites) for these purposes. Note that such materials have been overlooked so far as regards their energy storage potential, as indicated in our conclusions. In this work we provide a simple yet insightful model (see our Eqs. 1-7) strongly suggesting that such materials may present unique and highly tunable storage properties. $\text{Bi}_{1-x}\text{Nd}_x\text{FeO}_3$ is a flexible system that turns out to be an excellent case to support the materials design strategy (and simple model) we propose.

2. “It is not easy to estimate the breakdown strength of materials, since the breakdown almost always occurs with extrinsic mechanisms related with defects such as oxygen vacancies. However, the intrinsic breakdown strength of materials is generally known to inversely proportional to their electric bandgaps. Wang suggested an empirical formula for the relationship between intrinsic breakdown strength and bandgap of semiconductors and insulators. [Wang, proceedings of

25th conference on microelectronics, 2006] Based on the equation suggested by Wang, the intrinsic breakdown strength of $\text{Bi}_{1-x}\text{Nd}_x\text{FeO}_3$ (Bandgap was $\sim 2.7\text{ eV}$ for BiFeO_3 and $\sim 2.5\text{ eV}$ for NdFeO_3 , and 2.7 eV was taken.), PLZT ($\sim 3.4\text{ eV}$), and HZO ($\sim 5.7\text{ eV}$) were estimated as ~ 4.2 , ~ 8.4 , and 20.0 MV/cm , respectively. The maximum electric field used for energy storage were 2.6 , 3 , and 4.5 MV/cm for BiFeO_3 , PLZT, and HZO, respectively, which were ~ 62 , ~ 36 , and $\sim 23\%$ of intrinsic breakdown strength. The authors took a value from a literature, but it was from an extremely thin ($\sim 4.6\text{ nm}$) film for ferroelectric tunnel junction applications, and was measured using atomic force microscope. This value (62% of intrinsic breakdown field) was unreasonably high compared with those in other reports. Under such (unrealistic) assumption, the dielectric constant has significant impact on the calculated energy storage density ($\epsilon_0\chi_0 E^2/2$). The χ_0 values in table I of SI were in the range of 161 (Nd content of 0.4 and (001) direction) – 281 (Nd content of 0.4 along (110) direction). The energy storage density due to χ_0 values of 161 and 281 could be ~ 48 and $\sim 84\text{ J/cm}^3$ at an external field of 2.6 MV/cm which was used in the manuscript. These values were surprisingly large compared with the total energy storage density, meaning that a significant portion of the calculated energy storage density originated from dielectric response. When the external electric field is decreased to be $\sim 1.5\text{ MV/cm}$ (36% of intrinsic value as for the case of PLZT for fair comparison), the energy storage density due to χ_0 values of 161 and 281 decrease to ~ 16 (decrease by ~ 32) and ~ 28 (decrease by ~ 56) J/cm^3 . Moreover, when Nd content is higher than 70% , the field induced phase transition might not occur within the maximum field of $\sim 1.5\text{ MV/cm}$, so the energy storage density might decrease even more seriously. Inversely, let's assume that $\sim 62\%$ of intrinsic breakdown strength of HZO can be applied (12 MV/cm), then the energy storage density only from dielectric response (when χ_r is 30) could be as high as $\sim 204\text{ J/cm}^3$, which is already much larger than the largest value reported in this manuscript. Please note that this large value does not count on the field induced ferroelectric polarization value. In practice, even for the high quality HZO film, only $\sim 3\text{ MV/cm}$ can be reliably applied without significant breakdown.

To summarize, since the energy difference between polar and nonpolar phase increases with increasing Nd content, the energy storage density related with phase transition ($P_{\text{FE}}^0 E_{\text{down}}$) increases. However, when the practical breakdown strength is considered (far lower than the theoretical breakdown field), the field induced phase transition cannot be observed when Nd content is larger than 70% . For the case of the films with lower Nd content, the energy storage density due to dielectric response is rather high, but this value was too much overestimated due to the overestimated breakdown strength.”

Answer: We thank the Reviewer for his/her careful estimation of the breakdown fields. It is first important to realize that our main reasons to include in the manuscript results up to 10.8 MV/cm are that (i) a field of 6.5 MV/cm has been experimentally reported in BiFeO_3 (ACS Nano 7, 5385–5390 (2013)) and (ii) rare-earth substitution is also experimentally known to reduce the leakage current.

Moreover, our Fig. 5 of the manuscript provides the energy density and efficiency as a function of E_{\max} , implying that the storage performance at lower fields can also be found. In particular, within the maximum field of 2.6 MV/cm provided by Reviewer #1 for BFO, the energy density and efficiency of $\text{Bi}_{1-x}\text{Nd}_x\text{FeO}_3$ is still very promising. As a matter of fact, comparison with the experimental data on HZO (Adv. Energy Mater. **4**, 1400610 (2014)) and PLZT (Appl. Phys. Lett. **104**, 263902 (2014)) is shown in the figure above, and reveals that, with the maximum field of 2.6 MV/cm (or 2.81 MV/cm as reported in Appl. Phys. Lett. **100**, 062906 (2012)), W and η of BNFO are still considerably higher than those of HZO and PLZT (even if the W and η of HZO and PLZT are obtained with higher E_{\max} , such as 5 MV/cm). To address this point, we now include more experimental data in Fig. 5 of the manuscript, and add a corresponding discussion in the manuscript at the bottom of page 8.

Furthermore, we agree with the Reviewer that the dielectric constant has a sizable contribution to the total energy density, in particular for high electric field. In fact, this can be seen from the derived model (Eqs. 1-7), and is emphasized in the Conclusions of the manuscript.

Finally, we also show in Fig. S6 of the SI that, besides the sole effect of Nd composition, epitaxial strain is also predicted to be very effective to affect the P - E curves such that it allows to maximize the energy storage performance considering an achievable breakdown.

In short, we remain convinced that BNFO is promising for energy storage, and that the present manuscript is also important to attract attention to the study (and

understanding) of the many antipolar materials (such as *Pnma* perovskites) that have been overlooked so far for such purposes – which is in line with the following sentence of Reviewer #2: “It is of great interest that such a system could also play a potential role in the energy storage”.

Reviewer #2 (Remarks to the Author):

“In present study, the authors have proposed a lead-free antiferroelectric material for energy storage. For the material itself, the rare earth element substituted BiFeO₃ has been studied extensively. Most of them were focused on the crystal structure, ferroelectricity, and magnetism. It is of great interest that such a system could also play a potential role in the energy storage. The manuscript could be accepted for publication in NC.”

Answer: We thank this Reviewer for his/her positive assessment of our study.

“However, the authors need improve the manuscript by considering the following points.”

1. “Since the energy storage in antiferroelectric materials has been proposed for a while and many systems have been studied, could the authors provide a general picture on why Nd doped BiFeO₃ has such a superior performance? To do this, it is of great importance that the authors provide the simulation results of PbZrO₃ for comparison.”

Answer: This is a very good point. The difference in energy density between Bi_{1-x}Nd_xFeO₃ and PZO can be, e.g., understood from Eq. 1 of the manuscript. For that, taking the experimental *P-E* loops of PbZrO₃ (Appl. Phys. Lett. **105**, 112908 (2014) gives a critical field E_{down} of about 0.4-0.5 MV/cm and a polarization of the ferroelectric phase under *E*-field of about 50 $\mu\text{C}/\text{cm}^2$. In Bi_{1-x}Nd_xFeO₃, taking pure NdFeO₃ system under an electric field being along [001] as an example, E_{down} is computed here to be about 2 MV/cm, and the polarization of the T phase is about 80 $\mu\text{C}/\text{cm}^2$. Hence, the key difference is the much larger electric field needed to reach the polar phase in NdFeO₃ and, generally, in all anti-polar compositions of Bi_{1-x}Nd_xFeO₃; which is a consequence of the fact that the energy difference between the antipolar and polar phases in BNFO can be made relatively large as the Nd content grows, while the antipolar and polar phases of PZO are known to be virtually degenerate in energy [see e.g. Íñiguez et al., Phys. Rev. B 90, 220103(R) (2014)]. As a result, we have about 1 order of magnitude larger energy density in NdFeO₃ than in PZO.

Please also note that we do not currently have any effective Hamiltonian available for PZO, which currently prevents us from performing the types of calculations that were done for Bi_{1-x}Nd_xFeO₃ in the present manuscript. The main difficulty to

construct such an effective model for PZO is the existence of a very unusual coupling between cation displacements and oxygen octahedral tiltings (note that this coupling exists in the complex *Pbam* ground state of PZO, but vanishes in all the phases considered here for $\text{Bi}_{1-x}\text{Nd}_x\text{FeO}_3$), which requires non-trivial developments and adjustments of the models. We hope to be able to present the results for energy storage for PZO after we implement and test this new effective Hamiltonian scheme (i.e., probably in a year from now).

2. “In a previous paper (PRB 85, 064119 (2012)), one of the authors has revealed a complex phase diagram and morphotropic region in La-doped BiFeO_3 . In addition, it is noted that present simulation on Nd-doped BiFeO_3 is started from 60% BiFeO_3 . So the question is what is the exact phase diagram of $\text{Nd}_{1-x}\text{Bi}_x\text{FeO}_3$ and is the morphotropic region crucial for the performance in energy storage? Could it be considered in the simulation? As the authors described, compared with PZO, a three times increase of energy density was observed in BNLBTZ and was believed related to the morphotropic region)?”

Answer: Please note that we can indeed accurately compute properties of $\text{Nd}_{1-x}\text{Bi}_x\text{FeO}_3$ for various compositions, as evidenced by the fact that our calculated phase diagram of $\text{Bi}_{1-x}\text{Nd}_x\text{FeO}_3$ (Adv. Funct. Mater. 25, 552–558 (2015)) is in agreement with experiments (J. Mater. Sci. 44, 5102 (2009) and Phys. Rev. B 81, 020103 (2010)). In particular, we did find a morphotropic phase boundary (MPB), separating a rhombohedral *R3c* ground state for small Nd content from a *Pnma* ground state for larger Nd concentrations. We predicted this MPB in $\text{Bi}_{1-x}\text{Nd}_x\text{FeO}_3$ to be formed by complex nanotwinned phases that tend to display a (small) electrical polarization, which is detrimental for getting high energy density and efficiency. In contrast, it may be that the MPB of BNLBTZ involves the coexistence of multiple phases, which can lead to an enhancement of the dielectric constant that is advantageous for energy storage according to our Eq. (1). We have now added a section in the SI to discuss the effect of MPB on energy storage applications.

3. “Since Nd or La doped BiFeO_3 has been studied extensive, is there any experimental evidence (e.g. from P-E loop and dielectric measurement) that could support present theoretical results?”

Answer: These are valid points as well.

On page 8 of the SI and Figure S2 of the SI, the calculated *P-E* loop of $\text{Bi}_{1-x}\text{Nd}_x\text{FeO}_3$ with $x=0.1$ is compared with the corresponding experimental loop of Sm doped BFO (BSFO) with $x=0.06$ and $x=0.09$ (note that it is known experimentally that the *P-E* loops of rare-earth-doped BFO only depend on the average ionic radius, and that the average ionic radius of $\text{Bi}_{1-x}\text{Nd}_x\text{FeO}_3$ with $x=0.1$ is in-between those of BSFO with $x=0.06$ and $x=0.09$). Our predicted (rescaled) *P-E* loop is in rather good agreement with measurements.

Moreover, the measured dielectric susceptibility of ceramic NdFeO_3 extracted from Ref. 1 of the SI is about 180 at 323 K, which is comparable to our calculated values, i.e., 185 for E -field along [111], 209 for E -field along [001], and 233 for E -field along [110].

All these facts confirm the accuracy of the present calculations.

Reviewer #3 (Remarks to the Author):

“This manuscript reported energy storage properties of lead-free antiferroelectric $\text{Bi}_{1-x}\text{Nd}_x\text{FeO}_3$ system predicted based on calculation using a first-principles-based computational method. Subject matter discussed and results reported in this current manuscript are of great interest to the audiences of the Nature Communications who are seeking solutions to high density energy storage applications. Technical presentation is mostly comprehensive and conclusions are largely supported by simulation and calculation results. However, there are a couple of critical issues must be addressed. Therefore, this manuscript is recommended for revision. Questions and suggestions are highlighted below.”

Answer: We thank this Reviewer for his/her positive assessment of our study.

Critical issues:

1. “As it has been clearly stated in the Supplemental Information (SI) material, the framework of this manuscript is built on top of two approximations: (1) the dielectric response is independent of the magnitude of applied field and (2) the AFE to FE (and FE to AFE) transition is abrupt. It is okay to assume these approximations. But, they should be made clear in the main text. So that the audiences can make their own judgement as how reasonable those approximations are. The review believes that the first approximation is valid only small applied field. Under high field condition (in the FE region), the increase in polarization decrease with increasing applied field. Please provide a brief discussion in your revised manuscript.”

Answer: Please note that these two assumptions are only used for the derivation of the simple phenomenological model that we introduce to better understand the essence of our numerical simulations. In contrast, our numerical simulations are based on a first-principles-derived effective Hamiltonian that does not rely on any such assumptions. In fact, the effective Hamiltonian simulations can indeed yield a non-linear dielectric response and a field-dependent dielectric susceptibility. We have now added sentences in the manuscript on page 10 to clearly mention these two assumptions, and their validity.

2. “All the calculation and simulation are conducted with unipolar hysteresis loops rather than the widely used bipolar hysteresis loops, contrary to your simulation results shown in Fig. 3a and SI Fig. 3 which had clearly shown that the remanent polarization is nonzero. This implies that if you had used a bipolar hysteresis loop instead of the unipolar loop, you would have gotten increased area enclosed by the hysteresis loops, i.e. reduced efficiency. Please estimate how much an error can potentially be introduced by using unipolar hysteresis instead of bipolar hysteresis loops in your revised manuscript.”

Answer: This is a very interesting point. The change of behavior between unipolar and bipolar loop (or even additional cycles) may be due to the existence of defects or domains in grown samples, since, as demonstrated in the figure above (for a few representative cases) and in Fig. S4 of the SI, a bipolar hysteresis yields similar P - E curves for positive and negative electric fields, and additional cycles yield qualitatively the same results. As a matter of fact, beyond E_{up} (or below $-E_{\text{up}}$) the predicted curves of increasing and decreasing E -field are nearly identical. We have added a section in the SI (pages 12 and 13) and Fig. S5 of the SI to show the bipolar hysteresis loops.

Clarity issues:

3. “Page 1 line 14: "...for experimentally achievable electric fields." Please be specific. This statement is confusing to the reviewer's point of view. You may give a value or a range for the applied field. Also, is "conventional electrostatic capacitors" the same as "dielectric capacitors"?”

Answer: We have added, in the abstract, the explicit value of the experimental applied field. The corresponding sentence now reads:

“...for experimentally achievable electric field (6.5 MV cm⁻¹)”

Moreover, “conventional electrostatic capacitors” has indeed the same meaning as “dielectric capacitors” or “conventional dielectric capacitors”. As it is mentioned in Ref. 4 of the manuscript, “conventional” refers to the capacitors that are mainly made of dielectric ceramics or dielectric polymers. In addition, “electrostatic capacitors” are very often used with a similar meaning as “dielectric capacitors”, but may indicate broader kinds of dielectrics, e.g., linear dielectrics, ferroelectrics, relaxors, and antiferroelectrics. We now change the wording “conventional electrostatic capacitors” by “dielectric capacitors” in the abstract.

4. “Page 4: Fig. 1b appears not referenced in the main text of the manuscript. If so, please consider of moving this figure to SI.”

Answer: We thank the Reviewer for carefully finding the missing reference of Fig. 1b in the main text. We now reference it in the “Model and analysis” section on pages 9 and 10, as Fig. 1b visually illustrates the energies and barriers that appear in the equations.

5. “Page 6 line 4 from the bottom: "remnant" should be "remanent".”

Answer: We have now corrected this typo.

6. “Page 7 line 2: "...the PE curve of Fig. 3." What are the solid lines in the Figure? Please explain.”

Answer: The solid lines in Fig. 3 are *P-E* curves of Bi_{1-x}Nd_xFeO₃ with different Nd compositions, as denoted by the legend on the right side for each composition. We have added a sentence in the captions of Fig. 3 to clarify the meaning of the solid lines.

7. “Page 7 line 5: "...E_{max} value up to 10.8 MV/cm..." Are you sure that the approximations that this current manuscript based upon are still valid under this high electric field? If not, please consider of decreasing the upper limit of applied field in Figure 5. The efficiency data shown in Fig. 5 are questionable to the reviewer. If you had used bipolar hysteresis for the calculation. The charging curve will NOT start from zero. Rather, there will be a substantial enclosed area at

near the AFE (near zero field) region. And the enclosed area increase with increasing maximum applied field. Please reference, Hu et al., "Temperature-dependent energy storage properties of antiferroelectric PLZT thin films," Appl. Phys. Lett. 104 (2014) 263902 for experimental data."

Answer: We agree with the Reviewer that a field of 10.8 MV/cm is very high, and has not yet been achieved experimentally in BFO or BRFO (R=rare earth). As indicated in our answers to the comments #1 and #2 of Reviewer #1, it is experimentally reported that a field of 6.5 MV/cm can be achieved (ACS Nano 7, 5385–5390 (2013)) and that rare-earth substitution is also experimentally found to reduce the leakage current; therefore, we include in the manuscript results up to 10.8 MV/cm so that the predicted behavior and trends can be observed. Please also note that, as indicated in our answer to comment #2 of Reviewer #1, the storage performance as a function of E_{\max} is provided in Fig. 5, which allows to find the energy density and efficiency at lower breakdown.

Moreover, the calculated bipolar hysteresis loop is shown in the answer to comment #2 of Reviewer #3. Based on this loop, it appears that the aforementioned enlarged enclosed area with increasing maximum applied field is likely due the defects and domains in grown samples. We now show the bipolar hysteresis loops in Fig. S5 of the SI, and discuss them on pages 12 and 13 of the SI.

8. "Page 8 line 1: "...both W and η increase with the maximum field value..." see above comments."

Answer: Interestingly, the literature indicates that the dependence of efficiency with the maximum field can be material specific, e.g., η is not apparently decreased (above 85%) in SrTiO₃-substituted BiFeO₃ (J. Am. Ceram. Soc., **96** 2699 (2013)). As similar to the enlarged loop with increasing maximum field discussed in comment #7 of Reviewer #3, defects and domains in grown samples can play a role on the dependency of W and η on the maximum field. Here, in our manuscript, we study defect-free systems, and it is clear that both W and η increase with the maximum field value in this case (as also consistent with our Eqs (1) and (2)). It will thus be very interesting if experimentalists might check the influence of defects and domains on the dependency of both W and η on the maximum field.

9. "Page 8 line 5: "For a fixed...pure NFO). Please recheck the validity of this discussion."

Answer: Please note that this discussion is based on the results shown in Fig. 5b and 5c. Above 2 MV cm⁻¹ and for the composition of 0.5, the energy density of the [110] case is similar to that of the [001] case, while the efficiency is much higher.

10. "Page 9 line 6 from bottom: What does "energy surface" refer to here? Please explain."

Answer: “Energy surface” on page 9 or “energy landscape” on page 10, in a simplified picture, can be understood with the schematic illustration of Fig. 1b. The antiferroelectric and ferroelectric phases are local minima of the energy surface. In principle this “surface” is multi-dimensional, depending on all the degrees of freedom that are required to describe the AFE-FE transformation. In Fig. 1b, we use the most important order parameter, i.e., the polarization, to visually schematize the energy surface in one dimension. To clarify this issue, we have added in the text a reference to Fig. 1b following the phrase of “energy surface of an AFE material”.

Summary of changes (changes in the manuscript are shown in red in the text)

1. To answer comment #2 of Reviewer #1, we have modified Fig. 5 of the manuscript to include more experimental data of PLZT and HZO to facilitate the comparison at lower E -field, as well as added a sentence regarding this comparison at the bottom of page 8.
2. To answer comment #2 of Reviewer #2, we have added a section in the SI (page 15) to discuss the effect of MPB on energy storage applications.
3. To answer comment #1 of Reviewer #3, we have added a few sentences on page 10 of the manuscript to clarify the two assumptions adopted for the derivation of the simple model.
4. To answer comment #2 of Reviewer #3, we have added a section in the SI (page 12 and 13) and Fig. S5 of the SI to show the bipolar hysteresis loops.
5. To answer comment #3 of Reviewer #3, we have added the explicit value of the applied field in experiment (ACS Nano 7, 5385–5390 (2013)), and changed “conventional electrostatic capacitors” to “dielectric capacitors” in the abstract.
6. To answer comment #4 of Reviewer #3, we have added a reference of Fig. 1b on pages 9 and 10 of the main text.
7. To answer comment #5 of Reviewer #3, we have changed “remnant” to “remanent” on pages 3 and 6.
8. To answer comment #6 of Reviewer #3, we have added a sentence in the captions of Fig. 3 to clarify the meaning of the colored solid lines.
9. To answer comment #7 of Reviewer #3, we now show the bipolar hysteresis loops in Fig. S5 of the SI, and discuss them on pages 12 and 13 of the SI.

10. To answer comment #8 of Reviewer #3, we have added a reference to Fig. 1b following the phrase of “energy surface of an AFE material” on page 9 of the manuscript.
11. On page 8, the abbreviation of “PLZT” is introduced when it first appears in the manuscript.

Reviewers' Comments:

Reviewer #1 (Remarks to the Author):

The authors argued that the breakdown field for BiFeO₃ film can be as high as 2.6 MV/cm based on only one literature, but the reviewer cannot agree with that the previous study justify such high breakdown field. From Yamada et al.'s previous study, which is for the ferroelectric tunnel junction, the resistance of BiFeO₃ film before electroforming is about 2 GΩ, so the magnitude of current at -1V (~2.2 MV/cm) is ~0.5 nA, which looks quite small but it actually corresponds to a huge current density. The capacitor area can be calculated based on its reported diameter of 180 nm, and the resulting current density is ~2 A/cm², which is already too high for any energy storage application. Even worse, if we calculate the leakage current under higher field in on-state (<-2V), the current density might increase by ~10⁴ times. The 2.6 MV/cm that the authors assumed should corresponds to this on-state region, so the leakage current level under 2.6 MV/cm electric field could be ~10⁴ A/cm². Even for the off-state of the junction, the leakage current density is still of the order of ~10² A/cm². This means that this film can never work as an energy storage media because there will be actually almost no charge accumulation due to the very high leakage current, which is precisely what the reviewer has concerned in the previous review round. In fact, for the energy storage application, the electrode area must be much larger than this ferroelectric tunnel junction work, which inevitably induce higher chance of including the defects, making the situation even worse. Another problem is that, in the work of Yamada et al., they have to have very thin BFO film to form the tunnel junction, which in fact decreased the chance of inducing a sort of impact ionization and the chance for breakdown was low. If the film thickness is increased to decrease the leakage current, the voltage must be increased too to get the same electric field. Now, the problem is that the chance of incorporating the defects within a thicker film is higher than that in the thinner film, which is very well known in thin dielectric film area, making the chance of breakdown higher in thicker film.

The reviewer strongly believes that the unreasonably high electric field considered in this study significantly mislead readers with the too much overestimated energy storage density, because they did not take into consideration the leakage current and possible breakdown. Therefore, the reviewer still believes that the authors should use reasonable lower electric field values for their calculation, and in this case, the resulting energy storage density must be similar to or even lower than the previously reported experimental values from other materials. When the general non-negligible discrepancy between theory and experiments is considered, this work cannot justify its suitability for publication in Nature Communications.

Reviewer #2 (Remarks to the Author):

The authors have improved the manuscript significantly and I believe that current proposal of BiRFeO₃ as an energy storage material is very important for the application of perovskite multiferroics.

Reviewer #3 (Remarks to the Author):

This manuscript reported energy storage properties of lead-free antiferroelectric Bi_{1-x}Nd_xFeO₃ system predicted based on calculations using a first-principles-based computational method. Subject matter discussed and results reported in this current manuscript are new and of great interest to the audiences of the Nature Communications who are seeking solutions to high density energy storage applications. This revised manuscript successfully addressed all the concerned and answered all questions from the reviewers with regarding to the earlier version of the manuscript. Technical presentation is comprehensive and conclusions are supported by simulation and calculation results. This manuscript can be accepted for publication based on the reviewer's point of view.

Reviewer #1 (Remarks to the Author):

1. “The authors argued that the breakdown field for BiFeO₃ film can be as high as 2.6 MV/cm based on only one literature, but the reviewer cannot agree with that the previous study justify such high breakdown field.”

Answer: We understand the concern from Reviewer #1, and the possible difficulty of reaching high electric fields in experiment. However, please note that there are at least two experimental studies reporting the application of high electric fields in BiFeO₃ systems: there is the work of Yamada *et al.* demonstrating the application of a field of 6.5 MV/cm (i.e., much higher than 2.6 MV/cm) in ACS Nano **7**, 5385–5390 (2013), and there is also the work of Chen *et al.* in Appl. Phys. Lett. **100**, 062906 (2012) that studied BiFeO₃ films up to 2.81 MV/cm. Moreover, Reviewer #1 kindly pointed out to us in his/her first report that “*The maximum electric field used for energy storage were 2.6, 3, and 4.5 MV/cm for BiFeO₃, PLZT, and HZO, respectively, which were ~62, ~36, and ~23 % of intrinsic breakdown strength*”. As a result, it appears that a field of $2.6/0.62=4.2$ MV/cm should be achievable in BiFeO₃.

2. “From Yamada et al.’s previous study, which is for the ferroelectric tunnel junction, the resistance of BiFeO₃ film before electroforming is about 2 GΩ, so the magnitude of current at -1V (~2.2 MV/cm) is ~0.5 nA, which looks quite small but it actually corresponds to a huge current density. The capacitor area can be calculated based on its reported diameter of 180 nm, and the resulting current density is ~2 A/cm², which is already too high for any energy storage application. Even worse, if we calculate the leakage current under higher field in on-state (<-2V), the current density might increase by ~10⁴ times. The 2.6 MV/cm that the authors assumed should corresponds to this on-state region, so the leakage current level under 2.6 MV/cm electric field could be ~10⁴ A/cm². Even for the off-state of the junction, the leakage current density is still of the order of ~10² A/cm². This means that this film can never work as an energy storage media because there will be actually almost no charge accumulation due to the very high leakage current, which is precisely what the reviewer has concerned in the previous review round. In fact, for the energy storage application, the electrode area must be much larger than this ferroelectric tunnel junction work, which inevitably induce higher chance of including the defects, making the situation even worse. Another problem is that, in the work of Yamada et al., they have to have very thin BFO film to form the tunnel junction, which in fact decreased the chance of inducing a sort of impact ionization and the chance for breakdown was low. If the film thickness is increased to decrease the leakage current, the voltage must be increased too to get the same electric field. Now, the problem is that the chance of incorporating the defects within a thicker film is higher than that in the thinner

film, which is very well known in thin dielectric film area, making the chance of breakdown higher in thicker film.

The reviewer strongly believes that the unreasonably high electric field considered in this study significantly mislead readers with the too much overestimated energy storage density, because they did not take into consideration the leakage current and possible breakdown. Therefore, the reviewer still believes that the authors should use reasonable lower electric field values for their calculation, and in this case, the resulting energy storage density must be similar to or even lower than the previously reported experimental values from other materials.”

Answer: We thank Referee #1 for his/her comment on this healthy discussion and information on BFO, and agree that breakdown is a key factor in practice. However, we would like to kindly point out several important facts. For instance, the aforementioned field of 2.6 MV/cm is closer to the field of 2.81 MV/cm applied in Chen *et al.*, Appl. Phys. Lett. **100**, 062906 (2012) than to the field of 6.5 MV/cm applied in Yamada *et al.*, ACS Nano **7**, 5385–5390 (2013). Moreover, the work of Chen *et al.* does not concern ferroelectric tunnel junctions, but is done on a film grown on a SrTiO₃ substrate. Furthermore, as already indicated in our previous response as well as in our revised manuscript, we decided to focus on (Bi,Nd)FeO₃ rather than on pure BiFeO₃ because it is well-known that the insertion of rare-earth ions into BiFeO₃ reduces the leakage current very significantly. Finally, and as discussed in our previous response, we demonstrated that the energy density and efficiency of Bi_{1-x}Nd_xFeO₃ is still promising at fields achieved in Chen *et al.*, Appl. Phys. Lett. **100**, 062906 (2012), as compared with HZO and PLZT. For instance, comparison with the experimental data on HZO (Adv. Energy Mater. **4**, 1400610 (2014)) and PLZT (Appl. Phys. Lett. **104**, 263902 (2014)) reveals that, with the maximum field of 2.6 MV/cm (or 2.81 MV/cm as reported in Appl. Phys. Lett. **100**, 062906 (2012)), W and η of BNFO are still considerably higher than those of HZO and PLZT (even if the W and η of HZO and PLZT are obtained with higher E_{\max} , such as 5 MV/cm), as shown in Fig. 5 of the manuscript.

3. “When the general non-negligible discrepancy between theory and experiments is considered, this work cannot justify its suitability for publication in *Nature Communications*.”

Answer: As explained above, we do not believe that there is a “non-negligible discrepancy between theory and experiments”. We also would like to kindly emphasize that the objectives of the manuscript are (1) not only to report the promising storage properties of the specific compound Bi_{1-x}Nd_xFeO₃, but also to draw attention to the very promising possibility of using antipolar materials with the *Pnma* perovskite structure (that is, the most abundant type of perovskites) for these purposes; and (2) to provide a simple yet insightful model (see our Eqs. 1-7), strongly suggesting that such materials may present unique and highly tunable storage properties. As indicated in the manuscript and Supplementary Information, we also show evidence that the energy density and

efficiency of $\text{Bi}_{1-x}\text{Nd}_x\text{FeO}_3$ materials can also be further tuned by varying temperature and epitaxial strain, which really demonstrate that $\text{Bi}_{1-x}\text{Nd}_x\text{FeO}_3$ is a flexible system that turns out to be an excellent case to support the materials design strategy (and simple model) we propose.

We therefore humbly but strongly believe that the present work constitutes an important contribution to the problem of energy storage based on antiferroelectric materials, which is going to motivate further studies. Such belief is supported by statements of Reviewers #2 and #3, in their first and second reports, such as *“It is of great interest that such a system could also play a potential role in the energy storage”*; *“are of great interest to the audiences of the Nature Communications who are seeking solutions to high density energy storage applications”*; *“I believe that current proposal of BiRFeO_3 as an energy storage material is very important for the application of perovskite multiferroics”*; and *“Subject matter discussed and results reported in this current manuscript are new and of great interest to the audiences of the Nature Communications who are seeking solutions to high density energy storage applications”*.

Summary of changes (changes in the manuscript are shown in red in the text)

1. In order to address comments 1 and 2 of Reviewer #1, we have changed a previous sentence on page 7 of the manuscript as “noting that fields of 2.81 and 6.5 MV cm^{-1} have recently been applied experimentally to BFO films [13,30] (in particular, 2.81 MV cm^{-1} is achieved with a 54 nm-thick BFO film grown on a SrTiO_3 substrate, and 6.5 MV cm^{-1} is achieved with a 4.6 nm-thick BFO tunnel junction)”. For the same reason, we added the work of Chen *et al.*, Appl. Phys. Lett. **100**, 062906 (2012) as our new Ref. 30.
2. To further address comments 1 and 2 of Reviewer #1, we have also modified Fig. 5 (and its captions accordingly) by adding a dotted vertical line in each panel to indicate the electric field of 2.81 MV cm^{-1} that has been experimentally achieved in Ref. 30.
3. To address comment 3 of Reviewer #1, we have added a sentence on page 9 “...by varying temperature and epitaxial strain, in particular that the strong dependence of E_{up} and E_{down} with respect to strain can effectively accommodate different E_{max} that can be achieved in experiment.”.

Reviewers' Comments:

Reviewer #1 (Remarks to the Author):

During the previous two rounds of review, the reviewer strongly pointed out that the electric field taken for calculating energy storage was unreasonably high. However, the authors still argue that their electric field is reasonable based on only two literatures, instead of taking reasonable electric field value. Even in the literature they suggested, the leakage conduction characteristics cannot justify their high electric field value. Especially during the 2nd round of review, the reviewer kindly explained why one of the literature (by Yamada et al.) cannot support their argument. The energy storing capacitor is certainly different from ferroelectric tunnel junction in that reference. For energy storing capacitors, one should be much stricter for leakage current density, since the charges and resulting electrical energy cannot be stored with high leakage current density, which was the case for Yamada et al.'s work. In fact, they needed high leakage because that was their aim for current-based memory operation. In the other reference by Chen et al., the piezoelectric property of BiFeO₃ film was reported, and there was no concern about leakage conduction characteristics. From the authors' response, the reviewer was surprised to see that the authors may not know what intrinsic breakdown strength actually means. The breakdown of oxide materials is always induced by extrinsic mechanism related with defects. As a result, the intrinsic breakdown strength has never been achieved in experiment, but the authors argued that the intrinsic breakdown strength of 4.2 MV/cm can be achieved for BiFeO₃ in their response letter by citing the reviewer's comment. The reviewer still believes that the authors should take reasonable electric field value. If not, the reviewer is afraid that this manuscript should mislead many readers based on the seriously exaggerated energy storage density based on unreasonably high applicable electric field. The reviewer can agree that staying with the current applicable electric field might be more eye-catching, but it should not be a right manner for scientific papers. Therefore, the reviewer cannot recommend its publication in Nature Communications within current form.

Reviewer #4 (Remarks to the Author):

Regarding BFO, I have experience with this system. I think breakdown fields of the order of 1-3MV/cm are reasonable, if the film quality is good. I think the referee is being too practical in his/her expectation. On the other hand, I think the authors can be less dramatic in their conclusions. Perhaps they could tone down the over-ambitious claims about how this will change the world and instead focus on the fundamental physics of the phase transition that they think could lead to such large energy storage values.. I also think the authors are being too insistent on

pointing to references in which the breakdown values are measured under rather tight conditions (for example the few nm thick films). Perhaps they could make the paper work without such serious comparisons and just state that experimentalists will rise to this challenge of making perfect materials to prove their point?

Following the recommended changes, I would recommend this paper for publication since it is asking some questions of the materials system and is pushing experimentalists to improve the quality of the films in order to unravel such phenomena.

Reviewer #1 (Remarks to the Author):

1. “During the previous two rounds of review, the reviewer strongly pointed out that the electric field taken for calculating energy storage was unreasonably high. However, the authors still argue that their electric field is reasonable based on only two literatures, instead of taking reasonable electric field value. Even in the literature they suggested, the leakage conduction characteristics cannot justify their high electric field value. Especially during the 2nd round of review, the reviewer kindly explained why one of the literature (by Yamada et al.) cannot support their argument. The energy storing capacitor is certainly different from ferroelectric tunnel junction in that reference. For energy storing capacitors, one should be much stricter for leakage current density, since the charges and resulting electrical energy cannot be stored with high leakage current density, which was the case for Yamada et al.’s work. In fact, they needed high leakage because that was their aim for current-based memory operation. In the other reference by Chen et al., the piezoelectric property of BiFeO₃ film was reported, and there was no concern about leakage conduction characteristics. From the authors’ response, the reviewer was surprised to see that the authors may not know what intrinsic breakdown strength actually means. The breakdown of oxide materials is always induced by extrinsic mechanism related with defects. As a result, the intrinsic breakdown strength has never been achieved in experiment, but the authors argued that the intrinsic breakdown strength of 4.2 MV/cm can be achieved for BiFeO₃ in their response letter by citing the reviewer’s comment. The reviewer still believes that the authors should take reasonable electric field value. If not, the reviewer is afraid that this manuscript should mislead many readers based on the seriously exaggerated energy storage density based on unreasonably high applicable electric field. The reviewer can agree that staying with the current applicable electric field might be more eye-catching, but it should not be a right manner for scientific papers. Therefore, the reviewer cannot recommend its publication in Nature Communications within current form.”

Answer: We fully understand the concern of Reviewer #1, and also realize that the upper limit of breakdown field is an important factor to the energy storage performance. Please also note that Reviewer #4 indicated that “Regarding BFO, I have experience with this system. I think breakdown fields of the order of 1-3MV/cm are reasonable, if the film quality is good”. Such fact is also consistent with the value of 2.81 MV/cm indicated in the work of Chen *et al.* in Appl. Phys. Lett. **100**, 062906 (2012). Moreover, we can evaluate the *intrinsic* breakdown strength to be 4.37 MV/cm based on an empirical formula (Ref [30] of the revised manuscript) and the experimental band gap (2.74 eV from Ref. [29] of the revised manuscript).

As a result, we now *only* allow our electric fields to vary up to 4-5 MV/cm (rather than up to 10MV/cm as done before, see the previous and new versions of Fig. 5) and

particularly emphasize and discuss our results for the more reasonable fields ranging between 2 and 3 MV/cm. We also further emphasize in the Conclusions the needs for improving breakdown strength in experiments (via better quality of films or single crystals) for energy storage applications.

With all these changes, we are confident our revised manuscript will not be misleading. At the same time, we hope it will motivate experimental groups to produce even better films in order to achieve even better storage properties.

Reviewer #4 (Remarks to the Author):

1. “Regarding BFO, I have experience with this system. I think breakdown fields of the order of 1-3MV/cm are reasonable, if the film quality is good. I think the referee is being too practical in his/her expectation. On the other hand, I think the authors can be less dramatic in their conclusions. Perhaps they could tone down the over-ambitious claims about how this will change the world and instead focus on the fundamental physics of the phase transition that they think could lead to such large energy storage values. I also think the authors are being too insistent on pointing to references in which the breakdown values are measured under rather tight conditions (for example the few nm thick films). Perhaps they could make the paper work without such serious comparisons and just state that experimentalists will rise to this challenge of making perfect materials to prove their point?”

“Following the recommended changes, I would recommend this paper for publication since it is asking some questions of the materials system and is pushing experimentalists to improve the quality of the films in order to unravel such phenomena”

Answer: We thank Reviewer #4 for his/her positive appraisal of our work and valuable suggestions. Following these suggestions, we have removed results for very high fields (up to 10 MV/cm in the previous version), and avoided references to breakdown fields (of about 6.5 MV/cm) that most of the community would perceive as unrealistically high. Instead, in the revised manuscript we focus on the predictions for the energy storage performance of BNFO for fields below an estimated intrinsic breakdown strength of 4-5 MV/cm, and we especially pay close attention and discuss the results associated with the more accessible electric fields of 2-3 MV/cm. Then, in the conclusions we emphasize the need and motivation for improving the quality of films, so as to increase the magnitude of the breakdown fields and obtain even better storage properties.

Summary of changes (changes in the manuscript are shown in red in the text)

1. In order to address the comments of Reviewer #1 and Reviewer #4, we have made the following changes to the manuscript.
 - Abstract: the values of energy density and efficiency are now for electric field of 2-3 MV cm⁻¹; and the last sentence is changed as “A simple model is derived to describe the energy density and efficiency of a general AFE material, providing a framework to assess the storage properties and the effects of all the variations, such as doping, electric field magnitude and direction, epitaxial strain, temperature, etc., which can facilitate future search of AFE materials for energy storage.”.
 - Page 3, top paragraph: we have revised the storage properties by referring to the electric field of 2.81 MV cm⁻¹ and now the sentence is “it is predicted to exhibit an energy density of 109–143 J cm⁻³ for $x > 0.5$ and a maximum electric field of 2.81 MV cm⁻¹ (which has been experimentally achieved in BFO films) and a good efficiency (68-88%)”.
 - Page 7, bottom paragraph: we have revised the storage properties by limiting the applied electric field up to the estimated breakdown, and now the relevant sentences are “We considered E_{max} values up to 4.37 MV cm⁻¹ (dotted line in Fig. 5), which is the intrinsic breakdown field estimated based on the empirical expression of Ref. [30] and the experimental band gap³¹. Note that a field of 2.81 MV cm⁻¹ has recently been applied experimentally to BFO films¹³ (54 nm-thick BFO film grown on a SrTiO₃ substrate)”.
 - Page 8, second paragraph: “For a fixed Nd composition of 0.5, W for the [110] case is in overall comparable to that of [001] case over a wide range of E_{max} values” is now changed to “For a fixed Nd composition of 0.5, W for the [110] case is comparable to that of [001] case for E_{max} above 2 MV cm⁻¹”.
 - Page 8, bottom paragraph: since Fig. 5 is now limited to be below 5 MV cm⁻¹, the data of PVDF is not visible in the figure. Accordingly, PVDF is removed from the first sentence, and the comparison is now changed as “The energy density of BNFO is also much higher than that of PVDF (27 J cm⁻³ for $E_{max} \sim 8$ MV cm⁻¹)³³”.
 - Page 9, top paragraph: “for similar E_{max} ” is now changed as “for similar or lower E_{max} ”, and “e.g., close to 3 MV cm⁻¹” is changed as “e.g., 2-3 MV cm⁻¹”. And we revised the storage properties part as “Moreover, Fig. 5b further shows that, up to the estimated intrinsic breakdown field of $E_{max} = 4.37$ MV cm⁻¹, the energy density is predicted to be giant: it reaches values of 164, 191, and 213 J cm⁻³ for $x = 0.5, 0.7$ and 1, respectively, with the E-field along [001], the corresponding efficiency being large as well (76%, 88%, 91%, respectively). Similarly, both W and η are very large, i.e., 161 J cm⁻³ and 91%, for BNFO with $x = 0.5$ and E-field along [110] for the same E_{max} . These energy densities are comparable to that of supercapacitors (electrochemical capacitors), which is about 5 Wh kg⁻¹ (~ 125 J cm⁻³ with the mass density of BNFO)³⁷”.
 - Conclusions: “such as energy densities exceeding 250 J cm⁻³ and efficiencies above 90%, for amenable applied electric fields of the order of 6 MV cm⁻¹” is now changed as “such as energy densities of 150 J cm⁻³ and efficiencies of 88%,”.

for amenable applied electric fields of the order of 3 MV cm^{-1} . We also add a sentence at the end “Additionally, we hope our results will motivate the search for experimental strategies to push up the breakdown fields in these compounds, and thus move towards the superior storage properties that our simulations predict.”.

- Figure 5: the x -axis (E_{max}) is now up to 5 MV cm^{-1} , and the last sentence of the caption (about the dotted line) is changed as “The dotted vertical line denotes the estimated intrinsic breakdown field for BFO.”.

2. Other minor changes to the manuscript.

- Page 9, top paragraph: “applied in practice” is now changed as “achieved in practice”.
- Page 9, bottom paragraph: “in the SI we also demonstrates” is now changed as “in the SI we also demonstrate”.
- Page 13, bottom paragraph: “in particular that $Pnma$ phase is the most prevalent structure in perovskites” is now changed as “in particular the $Pnma$ phase which is the most common structure among perovskites”.